# ON ITERATIVE NEURAL NETWORK PRUNING, REINITIALIZATION, AND THE SIMILARITY OF MASKS

## ABSTRACT

We examine how recently documented, fundamental phenomena in deep learning models subject to pruning are affected by changes in the pruning procedure. Specifically, we analyze differences in the connectivity structure and learning dynamics of pruned models found through a set of common iterative pruning techniques, to address questions of uniqueness of trainable, high-sparsity subnetworks, and their dependence on the chosen pruning method. In convolutional layers, we document the emergence of structure induced by magnitude-based unstructured pruning in conjunction with weight rewinding that resembles the effects of structured pruning. We also show empirical evidence that weight stability can be automatically achieved through apposite pruning techniques.

## 1 INTRODUCTION

Deep neural architectures have seen a dramatic increase in size over the years (Amodei & Hernandez, 2018). Although not entirely understood, it is known that over-parametrized networks exhibit high generalization performance, with recent empirical evidence showing that the generalization gap tends to close with increased number of parameters (Zhang et al., 2017; Belkin et al., 2018; Hastie et al., 2019; Allen-Zhu et al., 2018; Du et al., 2018; Du & Lee, 2018), contrary to prior belief, also depending on the inductive bias and sensitivity to memorization of each base architecture (Zhang et al., 2019).

While advantageous under this point of view, the proliferation of parameters in neural architectures may induce adverse consequences. The computational cost to train some state-of-the-art models has raised the barrier to entry for many researchers hoping to contribute. Because of limited memory, time, and compute, and to enable private, secure, on-device computation, methods for model compression have seen a rise in popularity. Among these are techniques for model pruning, quantization, and distillation.

Pruning, in particular, has been seen as an overfitting avoidance method since the early decision tree literature (Breiman et al., 1984), with work such as that of Mingers (1989) comparing the effects of different tree pruning techniques. In analogy to this prior line of work, we provide an empirical comparison of pruning methods for neural networks. As explicitly pointed out in LeCun et al. (1990b), there is no obvious, natural order in which neural network parameters should be removed. Indeed, as Schaffer (1993) suggests, identifying the best network pruning technique should give way to *understanding* what accounts for their success.

Revived by the recent observation of the existence of sub-networks with favorable training properties within larger over-parametrized models ('lottery' or 'winning tickets' (Frankle & Carbin, 2018)), and in the spirit of Zhou et al. (2019), this work investigates the dependence of the properties of these sparse, lucky sub-networks on the choice of pruning technique. We set out to answer the questions: do pruning methods other than magnitude-based unstructured pruning give rise to winning tickets? If so, *do they identify the same lucky sub-network, or are there many equiperforming winning tickets within the same over-parametrized network?* This allows us to categorize pruning methods based on the similarity of masks they generate. We develop methods to further analyze the nature of trainable sparse networks found through common iterative pruning techniques, and provide new insight towards a deeper understanding of their properties and of the processes that generate them. Without addressing issues related to scalability and emergence of lottery tickets in

large-scale domains, we focus instead on the empirical characterization of weight evolution and emergence of distinctive connectivity patterns in small architectures, such as LeNet (LeCun et al., 1990a), on simple datasets, such as MNIST (LeCun et al., 1994). This also helps develop analysis techniques for sanity checks via visual inspection. We then apply the same techniques to larger architectures and more complex tasks to explore whether findings hold in different regimes.

## 1.1 CONTRIBUTIONS

This work provides empirical evidence showing that:

1. there can exist multiple different lucky sub-networks (lottery tickets) within an over-parametrized network;

2. it is possible to find a lucky sub-network through a variety of choices of pruning techniques;

3. lottery ticket-style weight rewinding, coupled with unstructured pruning, gives rise to connectivity patterns similar to the ones obtained with structured pruning along the input dimension, thus pointing to an input feature selection type of effect. This is not the case when rewinding is substituted by finetuning;

4. random structured pruning outperforms random unstructured pruning, meaning that networks are more resistant to the removal of random units/channels than to the removal of random individual connections;

5. different iterative pruning techniques learn vastly different functions of their input, and similarly performing networks make different mistakes on held-out test sets, hinting towards the utility of ensembling in this setting;

6. weight stability to pruning correlates with performance, and can be induced through the use of suitable pruning techniques, even without late resetting.

## 2 RELATED WORK

Identifying important units in a neural network is an open challenge in machine learning, driven by desires for interpretability and model reduction, among others. Various proxies for importance have been proposed as a direct consequence of the lack of precise mathematical definition of this term. Different objectives lead to different mathematical formulations of importance: concept creation, output attribution, conductance, saliency, activation, explainability, connectivity density, redundancy, noise robustness, stability, and more (Berglund et al., 2013; Amjad et al., 2018; Zhou et al., 2018; Dhamdhere et al., 2018; Shrikumar et al., 2017).

State-of-the-art results in the field have seemed to strongly correlate with increased number of parameters, leading to the common use of vastly over-parametrized models, compared to the complexity of the tasks at hand. In practice, however, most common neural network architectures can be dramatically pruned down in size (Hanson & Pratt, 1989; LeCun et al., 1990b; Hassibi & Stork, 1993; Han et al., 2015; Guo et al., 2016; Yang et al., 2016; Luo et al., 2017; Tung et al., 2017; Zhang et al., 2018; Ayinde et al., 2019; Mussay et al., 2019).

Unlike pruned pre-trained and then finetuned models, smaller untrained models are, in general, empirically harder to train from scratch. Frankle & Carbin (2018) conjecture that there exists at least one sparse sub-network, a *winning ticket*, within an over-parametrized network, that can achieve commensurate accuracy in commensurate training time with fewer parameters, if retrained from the same initialization used to find the pruning mask. To find this lucky sub-network, they iteratively prune the lowest magnitude unpruned connections after a fixed number of training iterations (LeCun et al., 1990b; Han et al., 2015). Interestingly, while pruned weights are set to zero, unpruned weights are *rewound* to their initialization values, prior to the next iteration of training and pruning.

Researchers have encountered difficulties in scaling this method to larger models. Liu et al. (2018) were unable to successfully train small sparse sub-networks found through this procedure, when starting from VGG-16 (Simonyan & Zisserman, 2014) and ResNet-50 (He et al., 2015) trained on CIFAR-10 (Krizhevsky, 2009), and instead observed that random initialization outperformed weight rewinding. Similarly, Gale et al. (2019) were unable to successfully find a sub-network

with these properties within ResNet-50 trained on ImageNet (Russakovsky et al., 2015) and Transformer (Vaswani et al., 2017) trained on WMT 2014 English-to-German[1].

Slight modifications to the original procedure have been proposed in order to favor the emergence of winning tickets in larger models and tasks. Zhou et al. (2019) challenge the idea that unstructured, magnitude-based pruning is necessary for the emergence of winning tickets. They instead prune weights based on their change in magnitude between initialization and the end of training, and reset the surviving weights to their sign at initialization times the standard deviation of all original weights in the layer. Frankle & Carbin (2018) note that larger models such as ResNet-18 and VGG-19 require selecting unimportant weights globally instead of layer by layer, and recommend the use of learning rate warmup. Frankle et al. (2019) further add that *late resetting* of the unpruned parameters to values achieved early in training appears to be superior to rewinding the weights all the way to their initial values. Morcos et al. (2019) confirm the findings on the efficacy of global pruning and late resetting.

Although hard to find in larger regimes, when found, lottery tickets have been shown to possess generalization properties that allow for their reuse in similar tasks, thus reducing the computational cost of finding task- and dataset-dependent sparse sub-networks (Morcos et al., 2019).

On the contrary, alternative approaches to neural architecture optimization have taken a constructionist approach, starting from a minimal set of units and connections, and growing the network by adding new components (Stanley & Miikkulainen, 2002; Xie et al., 2019). Finally, other methods combine both network growing and pruning, in analogy to the different phases of connection formation and suppression in the human brain (Floreano et al., 2008; Dai et al., 2019).

## 3 METHOD

All networks in this section are trained for 30 epochs using SGD with constant learning rate 0.01, batch size of 32, without explicit regularization. The pruning fraction per iteration is held constant at 20% of remaining connections/units per layer. All experiment and analysis code is publicly available[2].

### 3.1 PRUNING METHODS

This works explores a variety of pruning techniques that may differ along the following axes:

**Neuronal Importance Definition**: In magnitude-based pruning, units/connections are removed based on the magnitude of synaptic weights. Usually, low magnitude parameters are removed. As a (rarer) alternative, one can consider removing high magnitude weights instead (Zhou et al., 2019). Non-magnitude-based pruning techniques, instead, can be based, among others, on activations, gradients, or custom rules for neuronal importance.

**Local vs. global**: Local pruning consists of removing a fixed percentage of units/connections from each layer by comparing each unit/connection exclusively to the other units/connections in the layer. On the contrary, global pruning pools all parameters together across layers and selects a global fraction of them to prune. The latter is particularly beneficial in the presence of layers with unequal parameter distribution, by redistributing the pruning load more equitably. A middle-ground approach is to pool together only parameters belonging to layers of the same kind, to avoid mixing, say, convolutional and fully-connected layers.

**Unstructured vs. structured**: Unstructured pruning removes individual connections, while structured pruning removes entire units or channels. Note that structured pruning along the input axis is conceptually similar to input feature importance selection. Similarly, structured pruning along the output axis is analogous to output suppression.

In this work, we compare: magnitude-based $\{L_1, \text{random}\}$ unstructured (US), $\{L_1, L_2, L_{-\infty}, \text{random}\}$ structured (S), and hybrid pruning. The hybrid techniques consists of pruning convolutional

---

[1]Translation Task - ACL 2014 Ninth Workshop on Statistical Machine Translation (URL: `https://www.statmt.org/wmt14/translation-task.html`)

[2]in anonymized form, at `github.com/iclr-8dafb2ab/iterative-pruning-reinit`

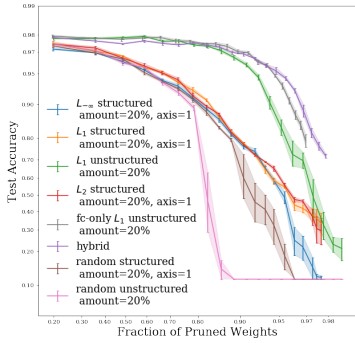 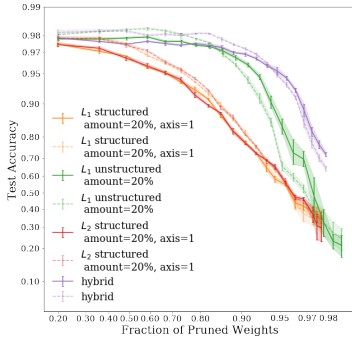

Figure 1: Test accuracy on the MNIST test set for SGD-trained LeNet models, pruned using six different pruning techniques, and rewound to initial weight values after each pruning iteration.

Figure 2: Test accuracy on the MNIST test set for SGD-trained LeNet models, pruned using four different pruning techniques corresponding to the four colors. The transparent curves correspond to finetuning; the dark ones to rewinding.

layers with $L_1$ structured pruning and fully-connected layers with $L_1$ unstructured pruning. "fc-only" identifies experiments in which only the fully-connected layers are pruned. Structured pruning is performed along the input axis. All techniques are local. We implement these pruning techniques and train our models using PyTorch (Paszke et al., 2017).

## 3.2 FINETUNING VS. REINITIALIZING

A point of contention in the literature revolves around the necessity to rewind weights after each pruning iteration, as opposed to simply finetuning the pruned model. Here we avoid performance-based arguments in favor of a study of how this choice affects the nature of left-over connectivity structure. For comparisons with alternative rewinding techniques, see Appendix A. Unless otherwise specified, all results refer to setup with full weight rewinding. Note that when rewinding the weights according to the strategies proposed in Frankle & Carbin (2018) and Zhou et al. (2019), all parameters in the network get affected by this procedure, including biases. On the other hand, in this work, pruning is *not* applied to biases.

## 4 RESULTS

In all figures, unless otherwise specified, the error bars and shaded envelopes correspond to one standard deviation (half up, half down) from the mean, over 5 experiments with seeds 0-4.

We begin by exploring the performance of sub-networks generated by different iterative pruning techniques starting from a base LeNet architecture, where lottery tickets are known to uncontroversially exist and be easy to find. Each point in Fig. 1 represents the test accuracy after exactly 30 epochs of training (not the best test accuracy achieved across the 30 epochs). Multiple techniques are able to identify trainable sub-networks up to high levels of sparsity. For a note on how the fraction of pruned weights is computed, see Appendix B.

Different types of magnitude-based structured pruning seem to perform only marginally better than pruning random channels, leading us to conclude that either the channels learn highly redundant transformations and are therefore equivalent under pruning, or there exists a hierarchy of importance among channels but it is not correlated to any of the $L_n$ norms tested in these experiments.

We quantify the overlap in sub-networks found by two different pruning methods started with pairwise-identical initializations by computing the Jaccard similarity coefficient (also known as Intersection over Union, or IoU) between the masks (Jaccard, 1901).

Although the different structured pruning techniques find sub-networks that perform similarly (Fig.1), a deeper investigation into the connectivity structure of the sub-networks they find shows,

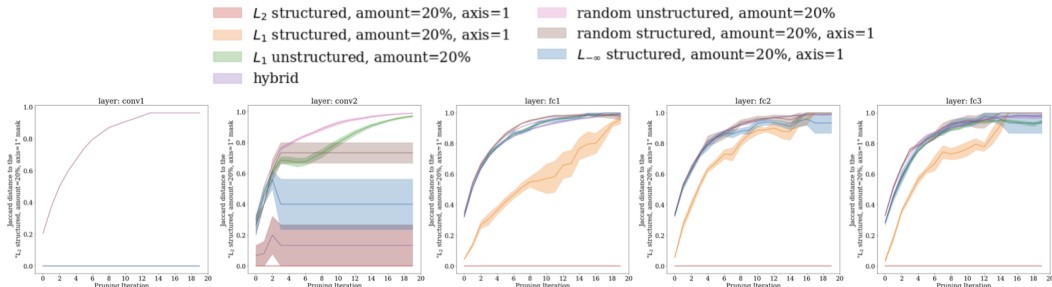

Figure 3: Jaccard distance between the masks (*i.e.* connectivity structures) found by pruning LeNet using $L_2$-structured pruning, and those found by other pruning methods listed in the legend, conditional on identical seed for meaningful comparison in light of neural network degeneracy. The comparison is conducted for each layer individually. $L_1$-structured pruning yields the most similar masks to $L_2$-structured pruning, as expected.

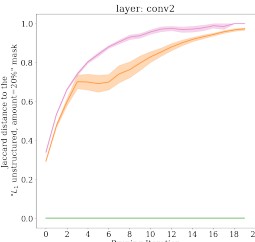

Figure 4: Jaccard distance between the mask for the second convolutional layer found by pruning LeNet using $L_1$-unstructured pruning, and the mask found by other pruning methods listed in the legend above, conditional on identical seed. Surprisingly, $L_1$-unstructured behaves more like $L_1$-structured pruning than a random unstructured pattern. This behavior is further investigated in Fig. 5.

as evident from the per-layer growth of the Jaccard distance with pruning iterations in Fig. 3, that there exist multiple lucky sub-networks with similar performance, yet little to no overlap.

The study of the Jaccard distance between masks surfaces another interesting phenomenon: as shown in Fig. 4, the connectivity patterns obtained from magnitude-based *un*structured pruning are relatively more similar to those that one would expect from structured pruning along the input dimension, especially in the convolutional layers, than to random unstructured patterns. This suggests that a form of input (or, at times, especially in larger models, output) feature selection is automatically being learned. This observation is further confirmed via visual inspection of the pruning masks; the first two columns of Fig. 5 show the weights and masks in the second convolutional layer of LeNet, obtained by applying structured and unstructured $L_1$ pruning. Unstructured pruning ends up automatically removing entire rows of filters corresponding to unimportant input channels. Structured pruning, instead, in this case, does so because, in these experiments, it is explicitly instructed to prune along the input dimension. This suggests that the low-magnitude weights pruned by $L_1$-unstructured pruning may be those that process low-importance hidden representations and whose outputs contribute the least to the network's output, which in turn results in lower gradients and smaller weight updates, causing these weights to remain small, and thus be subject to pruning.

Not only do the different pruning methods lead to different masks, but they also lead to sub-networks that learn partially complementary solutions, opening up the opportunity for the ensembling of different sparse sub-networks. Table 2 records the average accuracy (over the 5 experiment seeds) of sub-networks obtained at each pruning iteration through a set of pruning techniques, as well as the average accuracy obtained after simply averaging all their predictions, or the predictions of structured and unstructured $L_1$ pruning. Note that, after each pruning iterations, the levels of sparsity in the sub-networks will be different as they depends on the type of pruning applied.

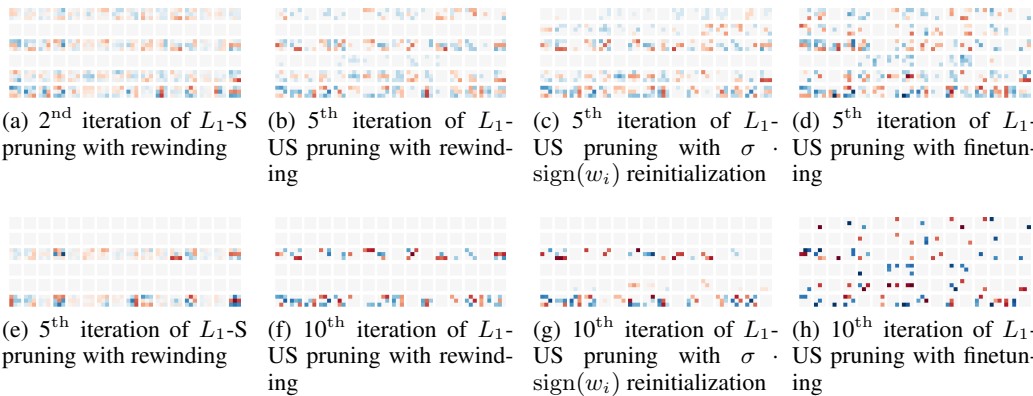

(a) $2^{nd}$ iteration of $L_1$-S pruning with rewinding

(b) $5^{th}$ iteration of $L_1$-US pruning with rewinding

(c) $5^{th}$ iteration of $L_1$-US pruning with $\sigma \cdot \text{sign}(w_i)$ reinitialization

(d) $5^{th}$ iteration of $L_1$-US pruning with finetuning

(e) $5^{th}$ iteration of $L_1$-S pruning with rewinding

(f) $10^{th}$ iteration of $L_1$-US pruning with rewinding

(g) $10^{th}$ iteration of $L_1$-US pruning with $\sigma \cdot \text{sign}(w_i)$ reinitialization

(h) $10^{th}$ iteration of $L_1$-US pruning with finetuning

Figure 5: Masked weights in the $2^{nd}$ convolutional layer of LeNet. The two rows represent two different pruning iterations. The columns represent four different pruning and weight treatments. This convolutional layer is composed of 6 input channels and 16 output channels, with $3x3$ filters. Masked out weights appear in gray color. Active, positive weights are depicted in red, negative weights in blue.

## 4.1 FINETUNING VS. REINITIALIZING

The "Lottery Ticket Hypothesis" Frankle & Carbin (2018) postulates that rewinding weights to the initial (or early-stage (Frankle et al., 2019)) values after pruning is key to identifying lucky sub-networks. In this section, we attempt to test this hypothesis in the experimental setting of small networks (LeNet) and simple tasks (MNIST), and we investigate, at the individual parameter level, how the this procedure ends up differing from finetuning after pruning.

The emphasis here is not on performance, rather on understanding. Therefore, we decide not to focus on extracting state-of-the-art performing lottery tickets, in favor, instead, of minimizing the influence of exogenous choices and the reliance on ad-hoc heuristics, whose primary goal is last-mile performance gains.

Finetuning is found to yield pruned networks that are comparable in performance with reinitialized networks (at least in the simple-task-small-network regime) when given the same training budget, and when the networks are pruned using the same technique. It is known, however, that higher quality lottery tickets can be obtained by rewinding the weight values to their value after a small number of training iterations (Frankle et al., 2019). This was not experimented with in this work.

When finetuning with $L_1$-unstructured pruning, the magnitude of weights tends to continue growing, achieving higher final values at later pruning iterations than their reinitialized counterparts.

Despite achieving similar average performance in this simplified scenario, the interesting observation lies in the stark difference between masks created by the same pruning technique when the weights are rewound or finetuned. Fig. 5 shows that finetuning removes the natural tendency of $L_1$-unstructured pruning with reinitialization to approximate the behavior of $L_1$-structured pruning. We can therefore primarily attribute the emergence of that interesting connectivity pattern to the choice of reinitializing the weights. In terms of mask structure, the difference between the effects of finetuning and lottery-ticket reinitialization is observed to grow similarly across all layers and logarithmically with the number of pruning iteration (see Appendix A).

## 5 WEIGHT EVOLUTION AND STABILITY TO PRUNING

Frankle et al. (2019) introduce the practice of late resetting, *i.e.* rewinding weights to their value after a small number of training iterations, as opposed to their value at initialization, and claim that its effectiveness is due to it enforcing a form of weight *stability to pruning*. In other words, weight values after a few batches of training may be sufficiently informative to then drive the optimization down the same promising direction, from which the high-quality solution found by the unpruned

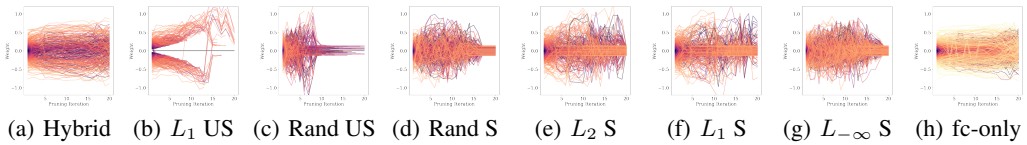

(a) Hybrid    (b) $L_1$ US    (c) Rand US    (d) Rand S    (e) $L_2$ S    (f) $L_1$ S    (g) $L_{-\infty}$ S    (h) fc-only

Figure 6: Weight values (y-axis) after 30 epochs of training at various consecutive sparsity levels (x-axis), for weights in the $2^{\text{nd}}$ convolutional layer in the LeNet architecture (seed: 0). Lines are terminated when the weight is pruned.

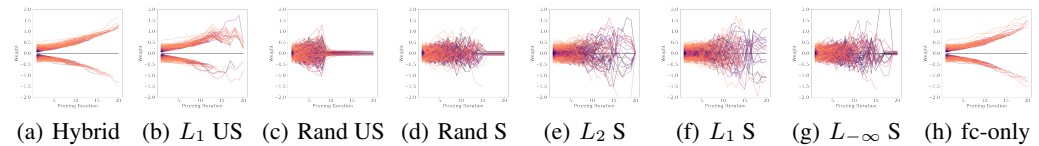

(a) Hybrid    (b) $L_1$ US    (c) Rand US    (d) Rand S    (e) $L_2$ S    (f) $L_1$ S    (g) $L_{-\infty}$ S    (h) fc-only

Figure 7: Weight values (y-axis) after 30 epochs of training at various consecutive sparsity levels (x-axis), for weights in the $3^{\text{rd}}$ fully-connected layer in the LeNet architecture (seed: 0). Lines are terminated when the weight is pruned.

model can be reached, even in high-sparsity scenarios. In practice, this would correspond to all unpruned weights converging to approximately the same value after each training stint.

While there exists a multitude of good solutions for convergence, the one initially found by the unpruned network is certainly a high-quality one that we know should be reachable from the initial conditions. In a way, being able to find this solution even in the sparse regime hints at the fact that the adopted pruning strategy has not interfered with and modified the optimization landscape to the point that this solution is no longer easy to reach.

We therefore ask whether weight stability to pruning is indeed advantageous, in terms of pruned network performance, and, if so, whether this property can be induced in more natural ways than by manual late resetting. To answer this, we track individual and group weight evolution over time in over-parameterized and under-parameterized networks (*i.e.* unpruned and pruned, with increasing sparsity). We look for differences in weight value after training when the weight is part of a large/small pool of unpruned weights.

As shown in Fig. 6, 7 and 8, we empirically find a correlation between weight stability and performance (see 1 for performance evaluation), and observe that the most competitive pruning techniques also exhibit the highest degree of weight stability, even when weights are reset to their true initial value (no late resetting).

In general, pruning techniques with an unstructured component tend to exhibit higher weight stability over pruning iterations. These weights tend to converge to solutions in which their ordinality varies only minimally from one pruning iteration to the next. This is especially true for the 'hybrid' pruning technique, in which weights in convolutional layers seem to converge to approximately the same value after each set of 30 allotted epochs of training from rewinding.

In fully-connected layers, even when using "stable" pruning techniques, the magnitude of unpruned weights at the end of training (with constant training budget of 30 epochs from the initial conditions) grows as sparsity grows, likely to compensate for the contribution of missing signal from pruned connections to the output of downstream units. As a parallel to the familiar decrease in the standard deviation of the weight distribution of common initialization schemes as the number of incoming and/or outgoing connections increases, here we confirm that, as we prune individual connections, the ideal, final weight configuration is composed of values with higher overall magnitude and higher standard deviation. We hypothesize, but do not get to test in this work, that, when pruning with rewinding, rescaling the initial weights by a factor that accounts for the decrease in the number of incoming and/or outgoing connections could be beneficial to achieve better, faster convergence in sparsified networks.

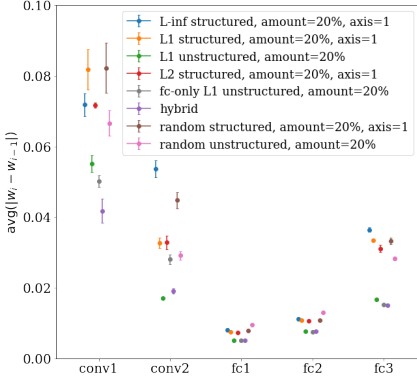

Figure 8: Absolute difference between weight values at pruning iteration $i$ and $i - 1$, averaged over all iterations and all weights in each layer. The lower, the more stable.

### 5.1 DO THESE OBSERVATIONS HOLD IN LARGER NETWORKS AND DOMAINS?

These empirical results have been observed to hold in AlexNet and VGG-11 architectures on MNIST and CIFAR-10 (see Appendix E). In even larger models, the nature of lottery tickets is still contended, and special care is required in designing ad-hoc training and pruning procedures to facilitate their discoverability. These larger regimes are not subject to study in this current work.

## 6 CONCLUSION

We show evidence against the uniqueness of winning tickets in a variety of networks and tasks, by identifying different lucky sub-networks of competitive performance within the same parent network, while controlling for degeneracy by fixing experimental seeds.

We also provide empirical results showing that rewinding weights to the original values at initialization after each pruning iteration yields sparsified networks that may not only be superior to finetuned sparse models from a performance perspective, but also, in practice, appear to possess structural advantages that might make them more suitable for hardware-level implementations with inference-time speed-up effects.

With the explicit intent not to adopt any special training tricks suggested in the literature to induce the emergence of lottery tickets in state-of-the-art deep architectures, this work intentionally restricts its action to smaller models and domains, to begin a more thorough investigation of the phenomenology of pruned networks.

We offer methods to experimentally analyze and compare the effects of different pruning techniques on the performance of neural networks and on the nature of the masks generated by each. This can be further extended in the future to better understand the role of late resetting, learning rate warm-up, and any new pruning method or related training trick, in terms of the similarity of masks obtained through each pruning procedure modification. We also conjecture that early monitoring of weight stability after a small number of pruning iterations (as opposed to, or in addition to the monitoring of the spectrum of the Hessian) might serve as a powerful indicator of the likelihood of the iterative pruning procedure to result in the discovery of a lottery ticket.

Valuable extensions would target machine learning applications beyond image classification, and study the interplay of different optimization strategies and pruning techniques in the evolution of weights and emergence of connectivity structure.

We hope these experimental results will guide theoretical work in this sub-field to match observations. An open dialog between the theoretical and experimental communities can lead to faster and more robust understanding of deep learning phenomena, and allow practitioners to select the most suitable methodology for their application based on more grounded understanding rather that mere performance.

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

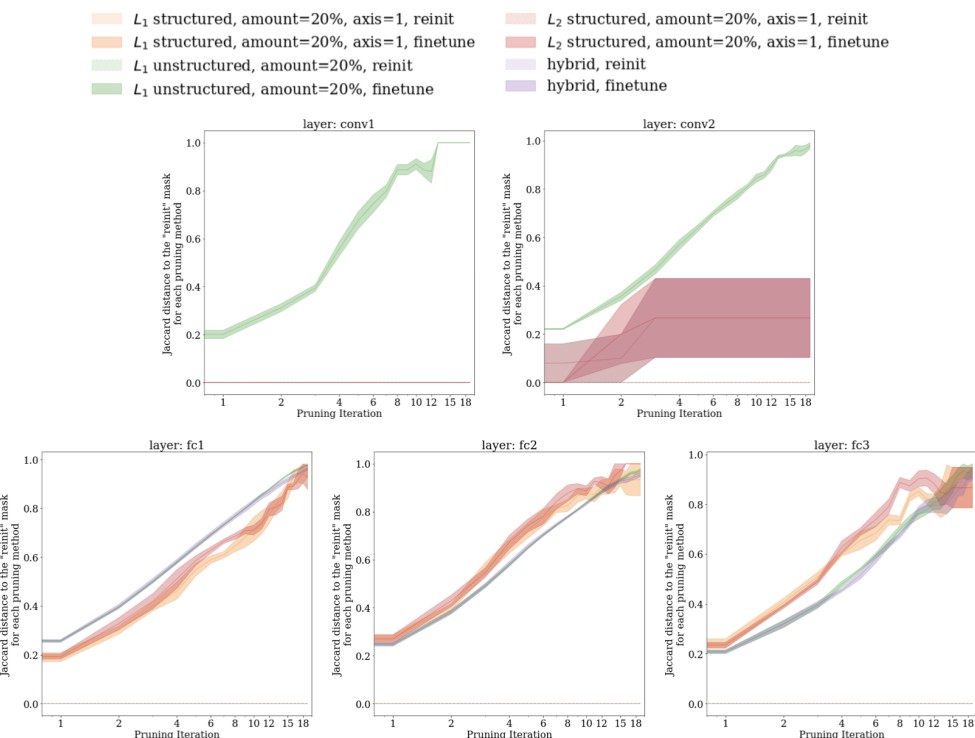

Figure 9: Jaccard distance between the masks found by pruning LeNet and rewinding weights, and those found by finetuning after pruning, conditional on identical pruning techniques and seeds. The comparison is conducted for each layer individually. Note the logarithmic scale on the x-axis.

## A   FINETUNING VS. REINITIALIZING (CONTINUED)

The Jaccard distance, or any other similarity measure among masks (*e.g.*, the Hamming distance), can be adopted to quantify the effects of the choice of finetuning or reinitializing weights after pruning. As expected, the nature of the connectivity structure that emerges in an iterative series of pruning and weight handling steps depends not only on the pruning choice but also on how weights are handled after pruning. Finetuning yields significantly different masks from rewinding, for all pruning techniques, and the difference (quantified in terms of the Jacccard distance) appears to grow logarithmically in the number of pruning iterations (Fig. 9).

### A.1   SIGN-BASED REINITIALIZION

*Does the exact weight value at initialization carry any significance, or is all we care about its sign?*

As proposed by Zhou et al. (2019), we experiment with reinitializing the pruned sub-network by resetting each unpruned parameter $i$ to $\sigma_{L_i} \cdot \text{sign}(w_i)$, where $w_i$ is the weight value of parameter $i$ at initialization, and $\sigma_{L_i}$ is the empirical standard deviation of the weights in layer $L$ that contains the parameter $i$. This is in contrast to rewinding to the initial weight values $w_i$, as originally suggested by Frankle & Carbin (2018), or finetuning without weight resetting.

The overall test accuracy of our experiments (Fig. 10) seems to support the observation of Zhou et al. (2019) that, as long as the sign matches the sign at initialization (*and* special care is taken in re-scaling the standard deviation), resetting the weights to different values won't affect the sub-networks trainability and performance. Removing the factor of $\sigma$ causes the network not to converge; as expected, keeping the weights within a numerically favorable range is important, and naively focusing on the sign alone leads to poor performance.

Adopting mask similarity to test for equivalence among reinitialization techniques, the method of Zhou et al. (2019) is found to induce structure that differs more substantially from the connectiv-

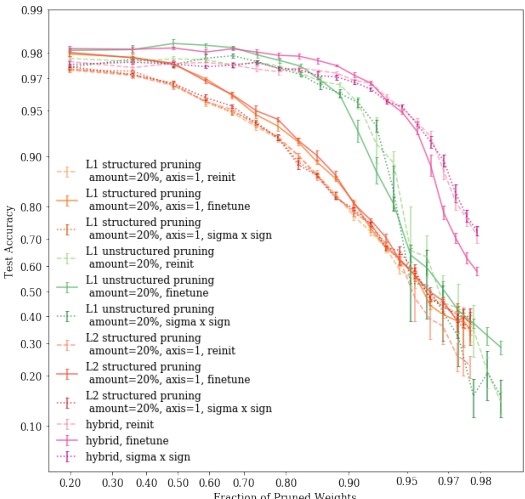

Figure 10: Test accuracy comparison on the MNIST test set for SGD-trained LeNet models, pruned using four different pruning techniques corresponding to the four colors, and reinitialized using three different techniques corresponding to the line styles.

|  | 0 | 0.2 | 0.4 | 0.5 | 0.6 | 0.7 | 0.8 | 0.9 | 0.95 | 0.97 | 0.98 | 0.99 |
|---|---|---|---|---|---|---|---|---|---|---|---|---|
| LeNet | 60 | 48 | 36 | 30 | 24 | 18 | 12 | 6 | 3 | 2 | 1 | 0.6 |
| AlexNet | 61,100 | 48,880 | 36,660 | 30,550 | 24,440 | 18,330 | 12,220 | 6,110 | 3,055 | 1,833 | 1,222 | 611 |
| VGG11 | 132,863 | 106,290 | 79,718 | 66,431 | 53,145 | 39,859 | 26,572 | 13,286 | 6,643 | 3,985 | 2,657 | 1,328 |

Table 1: Number of parameters (in thousands) left in each model at each pruning fraction.

ity patterns generated by the method of Frankle & Carbin (2018) than simple finetuning does, across most layers and a set of common choices of magnitude-based pruning techniques (Fig. 11).

## B    EFFECTIVE PRUNING RATE

A subtle implementation detail involves the way in which the fraction of pruned weights is computed, especially in convolutional layers pruned with structured pruning techniques. Take, for example, the LeNet architecture used in this work. When applying pruning to produce results like the ones displayed in Fig. 1, there are two ways to define the number of weights that get pruned (displayed along the x-axis). The definition used in the main portion of the paper uses the fraction of weights *explicitly* pruned by the decision rule that produces the mask for each layer. However, this doesn't account for the *implicitly* pruned weights, *i.e.* those weights that, due to downstream pruning in following layers, are now disconnected from the output of the neural network. For this reason, they receive no gradient and their value never changes from that at initialization. They can technically take on any value, including 0, without affecting the output of the neural network. In other words, they could effectively be pruned without any loss of performance. If we include these weights in the effective fraction of pruned weights, Fig. 1 (now converted into Fig. 12(a) for visual coherence) is modified into Fig. 12(b). When looking at effective sparsity, structure pruning appears to be competitive, especially at high pruning fractions, because its effective sparsity naturally tends to be higher. This also suggests that since the effective pruning rate per iteration is higher than 20% for structured pruning techniques, one could experiment with lowering it, to avoid aggressive pruning and consequent loss in performance.

The number of parameters corresponding to various fractions of pruned weights in the models investigated in this work is listed in Table 1.

## C    PRUNING AFTER 1 TRAINING EPOCH

*Is it necessary to train for many epochs before pruning to find high quality masks?*

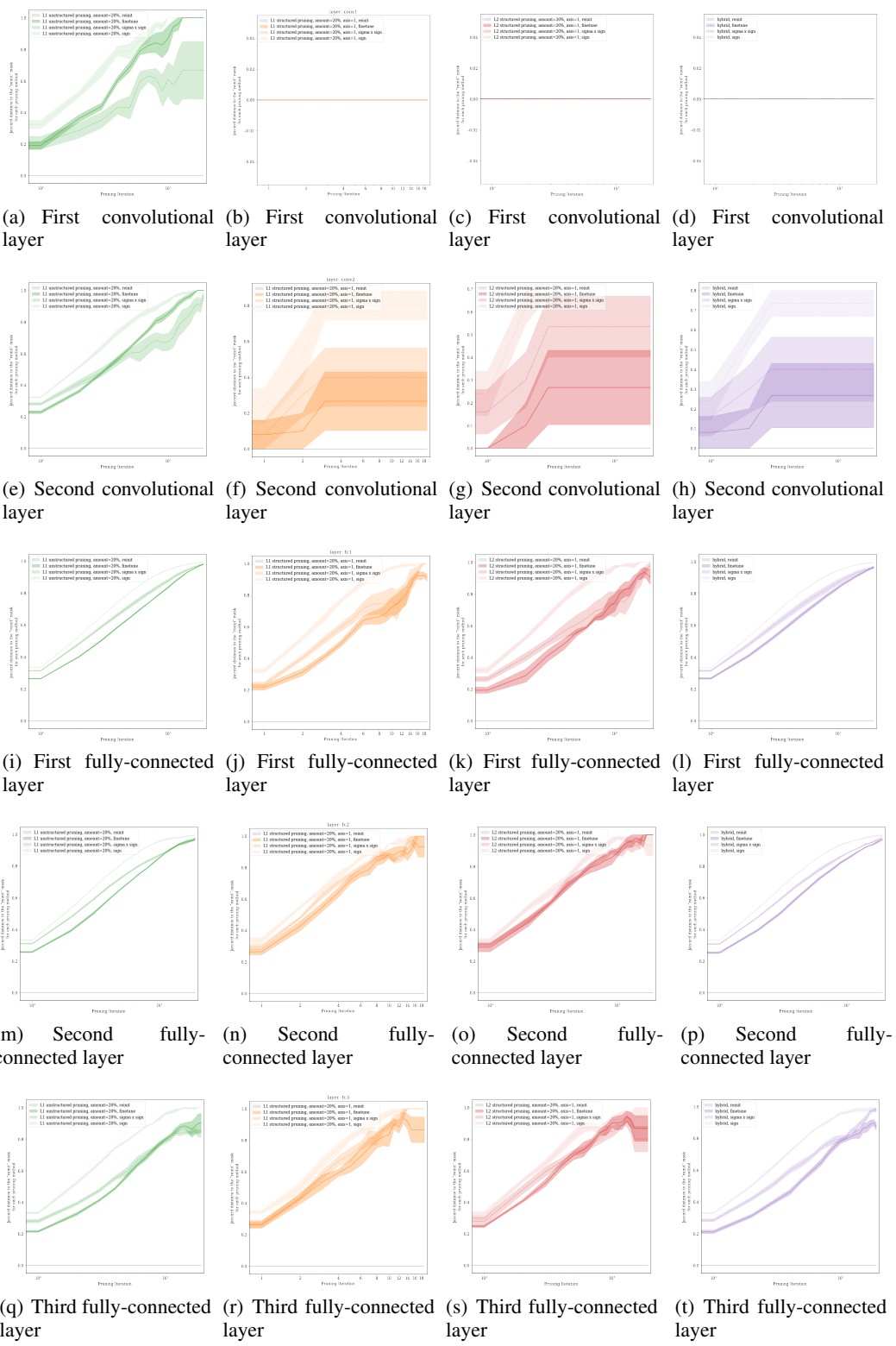

(a) First convolutional layer

(b) First convolutional layer

(c) First convolutional layer

(d) First convolutional layer

(e) Second convolutional layer

(f) Second convolutional layer

(g) Second convolutional layer

(h) Second convolutional layer

(i) First fully-connected layer

(j) First fully-connected layer

(k) First fully-connected layer

(l) First fully-connected layer

(m) Second fully-connected layer

(n) Second fully-connected layer

(o) Second fully-connected layer

(p) Second fully-connected layer

(q) Third fully-connected layer

(r) Third fully-connected layer

(s) Third fully-connected layer

(t) Third fully-connected layer

Figure 11: Growth of the Jaccard distance between the mask found by rewinding after pruning and three other techniques of handling weights (in order from highest to lowest opacity in the figures: finetuning, rewinding to $\sigma \cdot \text{sign}(w_i)$, and rewinding to $\text{sign}(w_i)$. Each column of sub-figures corresponds to a pruning technique (same color code as in the rest of the paper); each row corresponds to a layer in LeNet. Note the logarithmic scale on the x-axis.

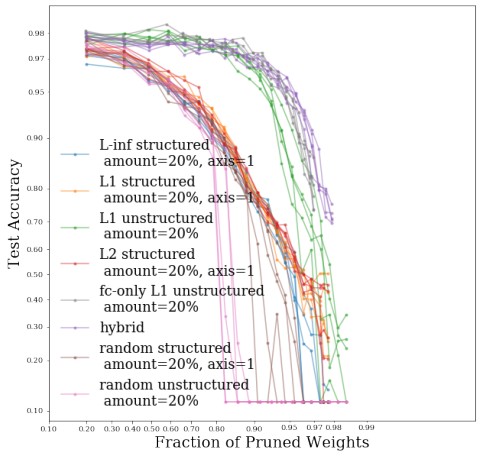 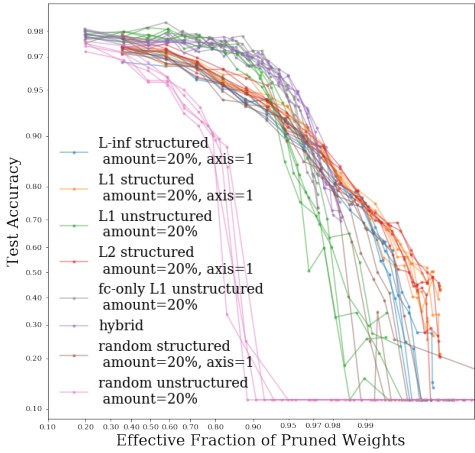

(a) Number of pruned weights corresponds to locations where the mask is explicitly zero.

(b) Unchanged units included in the effective number of pruned weights.

Figure 12: Test accuracy achieved after 30 training iterations by LeNet models trained with SGD and pruned with the methods listed in the legend. Each dot corresponds to a version of the model at the end of training. Each line connects models obtained by iterative pruning from the same seed. The two plots differ in the way the fraction of pruned weights is computed.

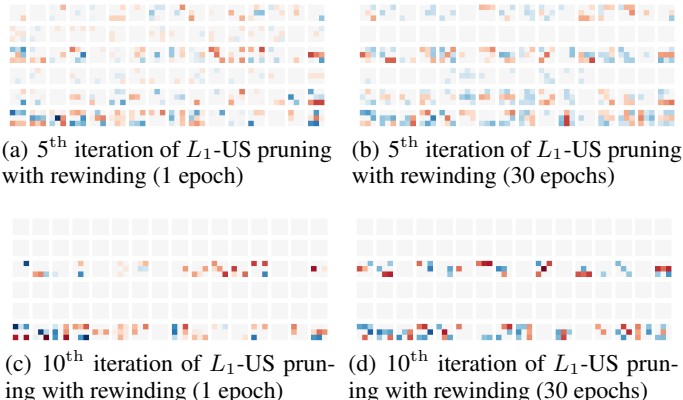

(a) $5^{\text{th}}$ iteration of $L_1$-US pruning with rewinding (1 epoch)

(b) $5^{\text{th}}$ iteration of $L_1$-US pruning with rewinding (30 epochs)

(c) $10^{\text{th}}$ iteration of $L_1$-US pruning with rewinding (1 epoch)

(d) $10^{\text{th}}$ iteration of $L_1$-US pruning with rewinding (30 epochs)

Figure 13: Difference in mask structure in the second convolutional layer that emerges when pruning after a single epoch (left) versus 30 epoch (right) of training (seed: 0).

To speed up the winning ticket search, we evaluate the option of shortening the training phases to only 1 epoch of training per iteration.

We compare the masks obtained with this quicker method to the masks found after training the network at each pruning iteration for 30 epochs (Fig. 13). After a single epoch of training, the ordinality of weight magnitudes has not yet settled to the solution corresponding to the values after 30 epochs of training. Although the majority of ranking swaps among weights, which can bring parameters from the lowest magnitude quartile all the way to the top one, and vice versa, happen in the first few epochs of training, minor movements later on in training might still be highly relevant in obtaining a robust ranking for magnitude-based pruning.

# D    COMPLEMENTARITY OF LEARNED SOLUTIONS

Pruning the same network using different pruning techniques gives rise to sparse sub-networks that differ not only in structure but also in the learned function that they compute. Given sufficient com-

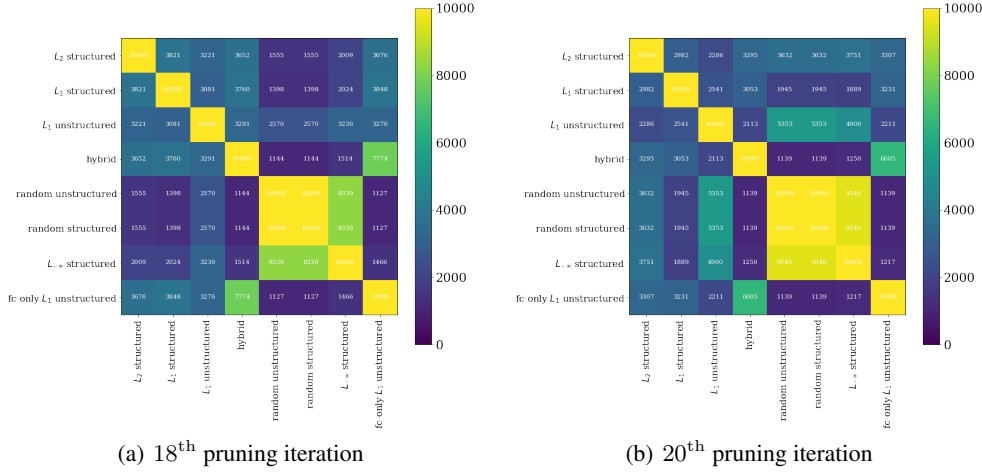

(a) $18^{\text{th}}$ pruning iteration    (b) $20^{\text{th}}$ pruning iteration

Figure 14: Number of examples in the MNIST test set over which the sub-networks obtained through each pruning technique agree on the prediction, on average (over 5 experimental seeds).

| Pruning Iteration | $L_2$ S | $L_1$ S | $L_1$ US | hybrid | random US | random S | $L_{-\infty}$ S | fc-only $L_1$ US | all | hybrid + fc-only + $L_1$ US |
|---|---|---|---|---|---|---|---|---|---|---|
| 1 | 97.4 | 97.5 | 97.8 | 97.8 | 97.4 | 97.3 | 97.1 | 97.9 | 98.2 | 98.1 |
| 2 | 97.2 | 97.0 | 97.7 | 97.6 | 97.0 | 96.9 | 96.9 | 97.8 | 98.2 | 97.9 |
| 3 | 96.5 | 96.6 | 97.8 | 97.5 | 96.1 | 96.4 | 96.2 | 97.7 | 98.1 | 97.9 |
| 4 | 95.8 | 95.6 | 97.9 | 97.6 | 95.6 | 95.1 | 95.4 | 97.8 | 97.9 | 97.9 |
| 5 | 95.0 | 95.1 | 97.6 | 97.5 | 93.8 | 94.0 | 94.1 | 97.8 | 97.7 | 97.9 |
| 6 | 94.2 | 93.8 | 97.6 | 97.4 | 92.2 | 92.9 | 93.3 | 97.6 | 97.6 | 97.7 |
| 7 | 91.7 | 92.7 | 97.4 | 97.5 | 89.2 | 91.0 | 91.3 | 97.6 | 97.4 | 97.8 |
| 8 | 89.5 | 90.9 | 97.3 | 97.4 | 45.8 | 88.2 | 88.9 | 97.5 | 97.2 | 97.7 |
| 9 | 87.6 | 86.9 | 97.0 | 97.3 | 14.0 | 83.7 | 86.2 | 97.5 | 97.2 | 97.5 |
| 10 | 82.0 | 82.2 | 96.5 | 96.9 | 11.3 | 79.2 | 81.3 | 97.3 | 96.8 | 97.3 |
| 11 | 77.2 | 77.7 | 95.9 | 96.8 | 11.3 | 60.1 | 76.3 | 97.0 | 96.6 | 97.1 |
| 12 | 72.5 | 72.5 | 94.5 | 96.4 | 11.3 | 45.5 | 74.1 | 96.9 | 96.3 | 97.0 |
| 13 | 69.0 | 65.1 | 90.2 | 95.8 | 11.3 | 41.1 | 65.6 | 96.1 | 95.7 | 96.4 |
| 14 | 65.4 | 58.3 | 83.5 | 95.3 | 11.3 | 32.0 | 58.4 | 95.9 | 95.1 | 96.3 |
| 15 | 55.7 | 54.7 | 72.7 | 94.4 | 11.3 | 18.5 | 48.3 | 94.7 | 94.6 | 95.7 |
| 16 | 47.1 | 43.7 | 69.5 | 92.5 | 11.3 | 11.3 | 24.7 | 93.3 | 94.0 | 94.7 |
| 17 | 45.9 | 41.6 | 47.9 | 88.3 | 11.3 | 11.3 | 21.9 | 90.7 | 91.9 | 92.8 |
| 18 | 36.7 | 37.8 | 32.4 | 81.8 | 11.3 | 11.3 | 14.5 | 87.4 | 91.0 | 91.6 |
| 19 | 30.2 | 35.9 | 23.0 | 76.4 | 11.3 | 11.3 | 11.9 | 84.3 | 89.3 | 90.3 |
| 20 | 29.4 | 33.1 | 21.2 | 71.9 | 11.3 | 11.3 | 11.7 | 78.6 | 85.0 | 86.0 |

Table 2: Sub-network accuracies at each pruning iteration. Ensembling of sub-networks obtained through different pruning techniques can yield higher performance, hinting at the complementarity of information learned by each sub-network.

pute and memory budget, one can consider combining the predictions made by each sub-network to boost performance. On the left side of Table 2, for each pruning iteration, the average accuracy of a pruned LeNet model is listed, along with, on the right, the accuracies obtained by simply averaging the predictions of all eight individual sub-networks ("all") and by averaging the predictions obtained from the three most promising pruning techniques (last column).

The similarity of solutions can also be explored by looking at the heat maps of the agreement in average class prediction across sub-networks obtained through different pruning techniques. For reference, Fig. 14 provides these visualizations for the $18^{\text{th}}$ and $20^{\text{th}}$ pruning iterations.

# E    ALEXNET AND VGG ON MNIST AND CIFAR-10

In this section, we confirm qualitative observations, previously reported on LeNet models, on the structure of connectivity patterns that emerge from the application of $L_1$ *un*structured pruning in the context of AlexNet and VGG-11 architectures.

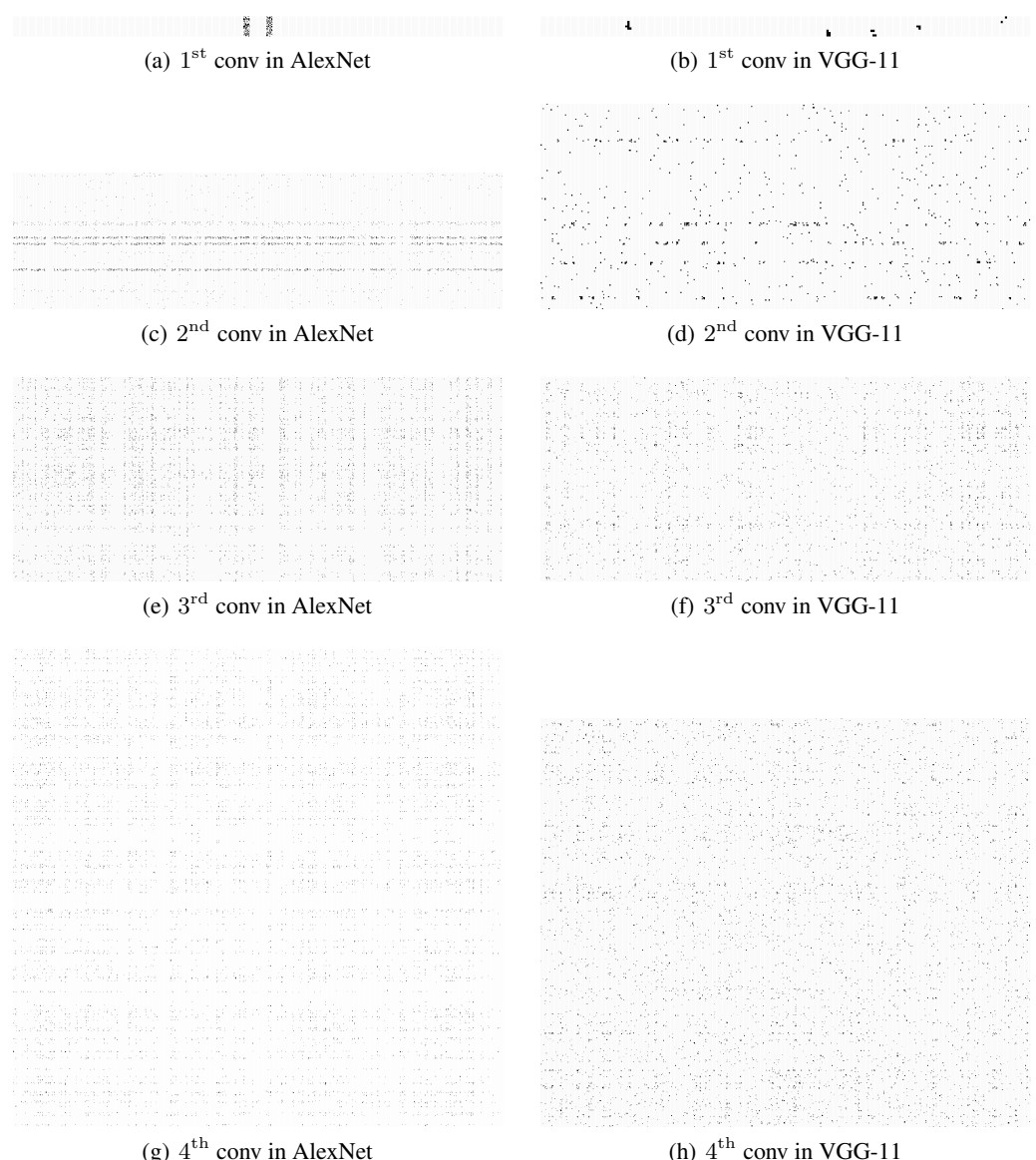

(a) $1^{st}$ conv in AlexNet

(b) $1^{st}$ conv in VGG-11

(c) $2^{nd}$ conv in AlexNet

(d) $2^{nd}$ conv in VGG-11

(e) $3^{rd}$ conv in AlexNet

(f) $3^{rd}$ conv in VGG-11

(g) $4^{th}$ conv in AlexNet

(h) $4^{th}$ conv in VGG-11

Figure 15: Binary weight masks for the first four convolutional layers of AlexNet trained on MNIST (left) and VGG-11 trained on CIFAR-10 (right), at the 20th and last pruning iteration for $L_1$ unstructured pruning (seed: 0). Despite the unstructured nature of the pruning technique, structure emerges along the input and output dimensions, which resembles the effect of structured pruning.

We train two individual sets of experiments starting from the base AlexNet model, one on MNIST, one on CIFAR-10. VGG models are trained on CIFAR-10 exclusively.

Fig. 15 shows the binary masks obtained in the first four convolutional layers of the two models after 20 pruning iterations, with pruning rate of 20% of remaining connections at each iteration. Here, the reported AlexNet properties refer to the MNIST-trained version. Preferential structure along the input and output dimensions (in the form of rows or columns of unpruned filters) is visible across the various layers, although visual inspection becomes hard and inefficient as the number of parameters per layer grows.

Qualitative visualizations of weight values at convergence over the series of 20 pruning iterations (Fig. 16) support arguments around weight stability as an indicator of the likelihood of identifying a competitive sub-network. In this small set of experiments, in which $L_1$ unstructured pruning

appears to remain competitive even at high sparsity levels, weights remain more likely to preserve their ordinality when subject to $L_1$ unstructured pruning. On the other hand, hybrid pruning, while exhibiting weight evolution characteristics that bridge the behavior of structured and unstructured pruning, is dominated by the effects of $L_1$ structured pruning of the convolutional layers. We suspect that too high a pruning fraction in the pruning techniques with a structured components may be responsible for the associated drop in performance. Regardless, the claim that weight stability is sufficient (though perhaps not necessary) for sub-network performance at high sparsity appears to hold in these architectures and tasks. As in previous experiments with smaller models, we want to shy away from performance-based comparisons that attempt to determine the "best" pruning technique, and instead reiterate that, irrespective of its value, performance seem to correlate positively with weight stability. However, this is not yet sufficient to determine a direct causal link between the two properties.

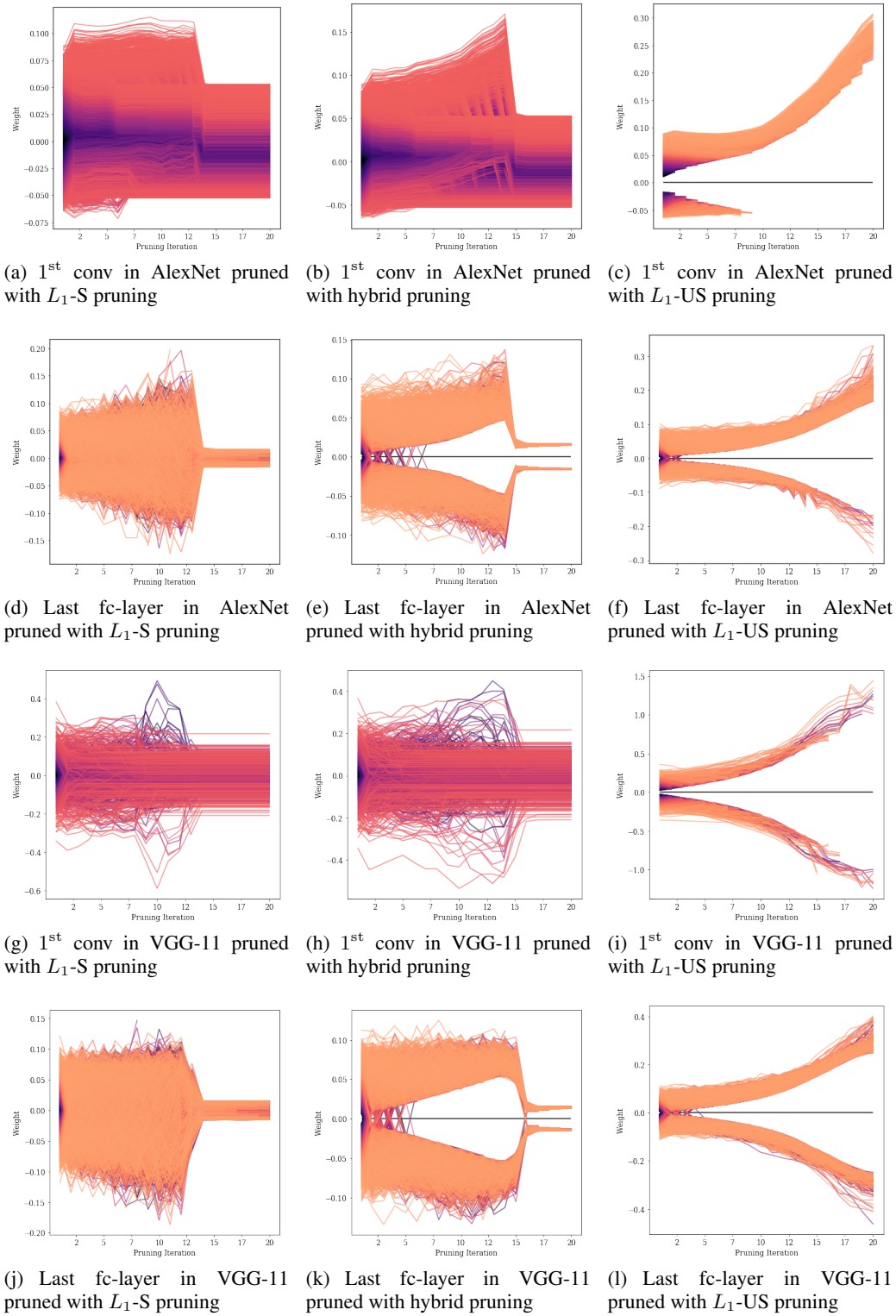

(a) $1^{st}$ conv in AlexNet pruned with $L_1$-S pruning

(b) $1^{st}$ conv in AlexNet pruned with hybrid pruning

(c) $1^{st}$ conv in AlexNet pruned with $L_1$-US pruning

(d) Last fc-layer in AlexNet pruned with $L_1$-S pruning

(e) Last fc-layer in AlexNet pruned with hybrid pruning

(f) Last fc-layer in AlexNet pruned with $L_1$-US pruning

(g) $1^{st}$ conv in VGG-11 pruned with $L_1$-S pruning

(h) $1^{st}$ conv in VGG-11 pruned with hybrid pruning

(i) $1^{st}$ conv in VGG-11 pruned with $L_1$-US pruning

(j) Last fc-layer in VGG-11 pruned with $L_1$-S pruning

(k) Last fc-layer in VGG-11 pruned with hybrid pruning

(l) Last fc-layer in VGG-11 pruned with $L_1$-US pruning

Figure 16: Weight values (y-axis) after 30 epochs of training at various consecutive sparsity levels (x-axis), for weights in the $1^{st}$ convolutional layer (first row) and last fully-connected layer (second row) in the AlexNet architecture, and the $1^{st}$ convolutional layer (third row) and last fully-connected layer (fourth row) in the VGG-11 architecture (seed: 0). Each column corresponds to one of the following pruning techniques: $L_1$ structured pruning, hybrid $L_1$ structured (in conv layers) and $L_1$ unstructured (in fully-connected layers) pruning, and $L_1$ unstructured pruning.

