# OpenReview forum: "On Iterative Neural Network Pruning, Reinitialization, and the Similarity of Masks"
_ICLR.cc/2020/Conference — Reject_

### Official Review · AnonReviewer1 · 2019-10-14
**Official Blind Review #1**

**Rating:** 1

**Review:**

There are major problems with this paper. It is concerned with the examination of pruning experiments for a LeNet on the MNIST dataset.  I fail to see how anything useful can be derived from this, as MNIST is a completely trivial dataset and LeNet is a very old, small architecture which does not at all resemble the massive overparameterised models that we care about.

From a narrative perspective, I am not sure what the key point is, what should the reader take home? What should they take account of when performing network pruning?

In terms of presentation, some of the figures are unreadable (figure 4). Figure 15 looks like noise. The writing is good however, if a bit grandiloquent.

I dislike writing short reviews, but I fear this paper falls too far short of ICLR standard.

Pros:
- Well written

Cons:
- Experiments are weak
- Unclear narrative; what's the one key message?

I have to give this paper a reject as the experiments conducted are far too weak, and there is little evidence anything found here will, say, generalise to a ResNet/DenseNet on ImageNet.



**Experience Assessment:**

I have read many papers in this area.

**Review Assessment: Checking Correctness Of Derivations And Theory:**

N/A

**Review Assessment: Checking Correctness Of Experiments:**

I assessed the sensibility of the experiments.

**Review Assessment: Thoroughness In Paper Reading:**

I made a quick assessment of this paper.

---

> ### Author Response · Authors · 2019-11-15
> **Rebuttal from the Authors**
>
> Thank you for reading our contribution.
>
> We would like to invite the reviewer to consider the strength and extent of experimentation performed in this paper by moving beyond the assessment of the choice of dataset and model. While we agree that one shouldn't claim that a proposed architecture achieves SoTA on "solved" (?) problems like MNIST, we would like to point out that the goal of this work is, in fact, _not_ to propose a model modification and use MNIST as a simple testbed to validate the performance (which we agree would be inconclusive). We maintain that MNIST is still an excellent dataset to study learning dynamics, weight co-adaptation, and deep phenomena in neural networks that we, as a field, are still far from understanding. We don't believe it advantageous to study these fundamental properties of neural networks (seen as physical objects with complex, perhaps chaotic dynamics) in large, complicated regimes and architectures when the simplest of cases (like MNIST with LeNet) is still just as poorly understood from the standpoint of the research being conducted in this paper (which, again, is not performance-oriented, with performance, instead, being a very well studied and thoroughly investigated property of this specific task).
>  We have performed a sensible search over pruning approaches of interest, and have documented nearly every decision along the way. We would like to encourage the reviewer to reconsider this point — scientific understanding of deep learning in its fundamental form will  need exhaustive experimental observation, and observational experiments are not definitionally weak.
>
> We believe the contributions main points are made very clear in section 1.1, titled contributions. We believe some of these are *directly* useful, such as #5 and #6, where #6 may help guide us towards designing stability induced procedures that may help with lottery tickets in larger models.
>
> Finally, observations on the small model-small dataset regime are incredibly important if we are to understand the minimum setting for these methods and approaches to work. We aim to shed light on some as-of-yet undocumented behaviors, and now the natural next step is to consider why they do or do not work in the large scale setting. We strongly emphasize that the purpose of this work is understanding, and we encourage the reviewer to reconsider the value of scientific, observational work rather than work that seeks to add modifications.

---

### Official Review · AnonReviewer2 · 2019-10-21
**Official Blind Review #2**

**Rating:** 3

**Review:**

This paper study the lottery ticket hypothesis by observing the properties of lottery tickets. In particular, the authors tested several different pruning techniques by varying evaluation criteria (L_1, L_2, L_-\infty and random) and pruning structures (structured, unstructured and hybrid). The authors perform experiments mainly on LeNet with the MNIST dataset and analyze the observations.

Overall, I think that the observations presented in the paper are not significant due to the following reasons.

First, the paper consists of the list of observations but how the observations extend to is not clearly described. There are no guidelines how to utilize the observations in future research (e.g., how they can be used for verifying the lottery ticket hypothesis or how they affect to existing pruning techniques) while some observations might be trivial or not very interesting (e.g., contribution 1 and contribution 2) for me.

Second, the observations are only presented for LeNet and MNIST and it is non-trivial whether they extend to large scale models. The authors present VGG11 and AlexNet results in Appendix but they are not large enough to verify their hypothesis for practice. The authors mentioned that larger models are not their subject, but this significantly reduces the confidence of the observations.

Other comments:
I think that Figure 5 is not well described. Explicitly noting the meaning of color in the figure would be better.

Texts in Figure 7 are too small to read.


**Experience Assessment:**

I have read many papers in this area.

**Review Assessment: Checking Correctness Of Derivations And Theory:**

N/A

**Review Assessment: Checking Correctness Of Experiments:**

I assessed the sensibility of the experiments.

**Review Assessment: Thoroughness In Paper Reading:**

I read the paper at least twice and used my best judgement in assessing the paper.

---

> ### Author Response · Authors · 2019-11-15
> **Rebuttal from the Authors**
>
> First of all, thank you for your comments.
>
> We would like to offer our point of view for why we disagree with the notion that the contributions and observations presented here are not interesting to the field. We agree that perhaps these approaches cannot directly be utilized at the moment to help reach SoTA on a given task. This utilitarian way of evaluating the contribution is at odds with the stated goal of the paper, which is to simply advance fundamental knowledge in the subdomain of science of deep learning. Many of the findings in this paper directly go to address major open questions around the nature and emergence of lottery tickets, including observations #1 and #2, which we therefore deem to be interesting and relevant to the field (or at least to those doing research in this sub-field). Objections to the absence of these studies have been raised in the community in the past to challenge the lottery ticket hypothesis itself. To the best of the authors knowledge, a thorough study of structure characterization of lottery tickets emerging from a multitude of pruning methods is itself of interest to better begin to understand more about this emergent behavior and move towards principled approaches to lottery ticket discovery.
>
> In addition, we disagree that observations on small models are not significant. If we are to understand the dynamics of what is happening in pruned models, under the lottery ticket hypothesis or any other hypothesis, we need to remove factors of variation introduced by SoTA seeking architectures. Even in the case where dynamics discovered in small networks do not apply to a large, say, ResNeXt or NasNet, that alone is interesting future work and important to understand and document. We do agree that confirmatory experiments in larger more complex domains would be a useful extension of this work, but not a necessary one to make these empirical discoveries worthwhile.
> While we agree that it is non-trivial to extend lottery tickets to larger models (as is well documented in the literature) we believe that understanding why and when lottery tickets emerge in smaller models will help us better apply them to larger models in the future.
>
> As per your direct comments, we have improved the description of Fig. 5.  The caption on Figure 7 already contains all the necessary information to decipher what the axes in the subplots represent (the numerical values are not important and the axes could be entirely removed in favor of simply showing the qualitative trend).

---

### Official Review · AnonReviewer3 · 2019-10-28
**Official Blind Review #3**

**Rating:** 3

**Review:**

*Summary*
This paper compares network pruning masks learned via different iterative pruning methods. Experiments on LeNet + MNIST show (a) different methods can achieve similar accuracy, (b) pruned sub-networks may differ significantly despite identical initialization, (c) weight reinitialization between pruning iterations yields more structured convolutional layer pruning than not reinitializing, and (d) pruning methods may differ in the stability of weights over pruning iterations.

*Rating*
There are interesting bits of data in this paper, but the overall story is somewhat muddled and some inferences seem to be insufficiently supported by data (1-2 below). In addition, the text would benefit from better organization and presentation (3-4 below) and replications on other datasets and architectures (5 below). As a result, my rating is currently weak reject.

(1) *Overlap in pruned sub-networks*: In the middle of Sec. 4, Fig 3-5 examine the similarity of pruning masks between methods. It seems clear from several of the plots that multiple methods produce identical layer-wise masks, e.g. Fig 3(a), while others show a wide variance. The overlap in lines makes this difficult to assess at times: perhaps a table would communicate it better? Also, are Fig 3-4 depicting the Jaccard distance between masks of unpruned or pruned weights? Is the ordering of training samples fixed in addition to network initialization? Is reinitialization used between iterations? Also, Fig 5 seems to contradict the conclusion that methods tend to learn different masks, since the structures are noticeably similar.

(2) *Weight stability during pruning*: It is difficult to discern a conclusion in Sec 5. First, a clarification on the figures: are lines for pruned weights terminated where they are pruned? If so, this would be helpful to state. The 4th paragraph claims, "we empirically find a correlation between weight stability and performance", but this is not at all obvious from Figures 6-7. I'm not sure what a more stable evolution looks like. Hybrid is shown to be accurate in Fig 1, but the conv. weights in 6(a) are a spaghetti tangle and the FC weights in 7(a) are constantly increasing in magnitude. Perhaps a mathematical formulation for stability (perhaps based on average standard deviation of each weight's values over training) with a table of values for each method/layer would help to clarify.

(3) *Organization*: Since the paper has many intertwined observations, a better organization would be helpful. Consider mirroring the structure of Sec 1.1 in a combined Sec. 4-5 with clear paragraph headers summarizing each conclusion.

(4) *Presentation*: Figure is too small throughout to read from a printed copy (or even on a screen without significant zooming). Several results could be presented with less ambiguity in tabular form, as noted above.

(5) *Replications*: The paper presents results only a single set of experiments using the MNIST dataset with the LeNet architecture. While this isn't a fatal issue, it is a significant weakness.

*Notes*
Fig 1 and 2: What spacing is used for the x- and y- axes?
Fig 8: Perhaps scale vertically by the standard deviation of the weights?

**Experience Assessment:**

I have published one or two papers in this area.

**Review Assessment: Checking Correctness Of Derivations And Theory:**

N/A

**Review Assessment: Checking Correctness Of Experiments:**

I assessed the sensibility of the experiments.

**Review Assessment: Thoroughness In Paper Reading:**

I read the paper at least twice and used my best judgement in assessing the paper.

---

> ### Author Response · Authors · 2019-11-15
> **Rebuttal from the Authors**
>
> We would like to thank the reviewer for their helpful and detailed comments!
>
> Overall, we thank the reviewer for considering the merits of purely observational work by itself — this is critical for moving with a scientific basis of understanding. Organizationally we believe that by stating the observations up front in Sec 1.1, we are able to lay out the story of our observations in the manner by which they were uncovered, an ordering and structure we feel to be more natural (related to point 3).
>
> We will answer and respond to specific questions/comments:
>
> (1) No method produces identical layer-wise masks to another, unless the layer is too small to be pruned at all (see conv1 for structured pruning). In all other cases, the line at distance = 0 is the baseline, and it's shown for sanity check.
> We believe that the graphical form is useful from an evolution standpoint — we note the curvature for the Jaccard distance when plotted against the pruning iteration is directly insightful. We agree that perhaps this visualization is not perfect, but, when representing it in a table, we found the information even harder to process and engage with, without immediate visual assistance. As the captions and plot labels clearly state, the Jaccard distance is computed between masks, not weights. The ordering of training samples is fixed (on top of the initialization). We tried to be as thorough as possible to control for any type of confounding factors and sources of variability that would not be directly caused by the effect we were trying to measure, i.e. the role of the pruning method. As per the lottery ticket procedure, and as we had hinted at in Sec.3.2, lottery tickets are searched for using rewinding to the initial weight values. Again, at first, to be able to focus in on the role of the choice of pruning technique, we conducted our experiments without varying any of the other knobs (the choice of reinitialization strategy being one of them). We did explore that dimension of variation as well, though, in the experiments in Appendix A. We did realize, thanks to your question, that Sec.3.2 was not entirely clear about this point, so we slightly modified the language there. In Figure 5, columns 2, 3, and 4 all use the same pruning technique (the difference here is only the reinitialization technique). The meaningful comparison is between column 1 and 2. Indeed, in the case of this specific seed and due to the very small size of the conv1 layer in LeNet, the masks do end up looking similar. For other seeds, instead, for example, although unstructured pruning continues to show structured-like patterns, the channels that end up getting pruned are _not_ the same ones that L1-structured pruning prunes. The per-layer distances are even more striking in larger, non-convolutional layers.
> (2) Yes, lines for pruned weights terminated where they are pruned; we now state that more clearly. The point of this section ("we empirically find a correlation between weight stability and performance") is made clearer when also considering the performance plot in Fig 1. We have added a note that makes this point clearer, also encouraging the reader to look at Fig 1 for a reminder. Regarding your later points, although we note that the evolution looks quite tangled in 6(a), in fact the weight magnitude per weight is not changing drastically from iteration to iteration and, more important, there doesn't seem to be much crossing from negative to positive, which had been identified in previous work as potentially key to the formation of lottery tickets. The same holds for 7(a), as from iteration to iteration there is little noise. We include a definition of stability and show the results you hint at in Figure 8 (see y-label). If the reviewers believe that Figure 8 is sufficient to illustrate the point and Figs 6-7 only confuse the reader, we'd be happy to remove them.
>
> (5) We present results on LeNet+MNIST for ease of interpretation — many of the phenomena we document here are difficult to reason about in larger models (though we agree that this will be very important in the future!). Our extended results (contained in the appendix) confirm some critical observations on larger scale models. We believe that large experiments would detract from the main points of the work at this time, and is welcome future work. It is known, as stated in the text, that lottery tickets are harder to find in larger domains and require the introduction of tricks that would introduce confounders in our experiments and invalidate the experimental setup.
>
> Notes:
>
> - The axes for Fig 1 and 2 use the "logit" scale setting in matplotlib. We found this to be the most appealing representation for the data we were plotting.
> - Scaling by the std deviation in this case does not change the comparative argument between methods, and we believe it would make the metric/value harder to reason about.

---

### Author Response · Authors · 2019-11-15
**Empirical Scientific Contributions in Machine Learning**

Dear area chairs, reviewers, and readers,

Thank you all for taking the time to read our contribution. We have answered concerns and questions at the individual reviewer level.

On the whole, the authors would like to argue, from our point of view, that the point of science is not to always be directly and immediately "useful" (in a SoTA sense) and that that might not be the correct lens to apply when assessing the merits or shortcomings of this work. While we strongly believe many of our contributions to be, indeed, useful (for example, that lottery tickets emerge from different strategies, or that there exists evidence of structure forming when unstructured pruning is used with rewinding -- which can inform ML engineers with inference-time concerns and hardware constraints in their choices of pruning strategy) we encourage the reviewers to consider the fact that purely observational work with well documented experiments (such as those presented in this work) alone constitute a valuable scientific contribution worthy of discussion at a conference, and capable of sparking new development in the research community. We are a long way from developing a principled understanding of deep emergent phenomena, and we believe this empirical work can successfully complement much of the theoretical work being carried out in this area.

---

### Decision · Program_Chairs · 2019-12-19

**Decision:**

Reject

**Comment:**

This is an observational work with experiments for comparing iterative pruning methods.

I agree with the main concerns of all reviewers:

(a) Experimental setups are of too small-scale or with easy datasets, so hard to believe they would generalize for other settings, e.g., large-scale residual networks. This aspect is very important as this is an observational paper.
(b) The main take-home contribution/message is weak considering the high-standard of ICLR.

Hence, I recommend rejection.

I would encourage the authors to consider the above concerns as it could yield a valuable contribution.